# Intact-Cell MALDI-ToF Mass Spectrometry for the Authentication of Drug-Adapted Cancer Cell Lines

**DOI:** 10.3390/cells8101194

**Published:** 2019-10-02

**Authors:** Jane F. Povey, Emily Saintas, Adewale V. Aderemi, Florian Rothweiler, Richard Zehner, Wilhelm G. Dirks, Jindrich Cinatl, Andrew J. Racher, Mark N. Wass, C. Mark Smales, Martin Michaelis

**Affiliations:** 1Industry Biotechnology Centre and School of Biosciences, University of Kent, Canterbury CT2 7NJ, UK; J.Povey@kent.ac.uk (J.F.P.); esaintas@icloud.com (E.S.); walerem@yahoo.com (A.V.A.); m.n.wass@kent.ac.uk (M.N.W.); 2Institut für Medizinische Virologie, Klinikum der Goethe-Universität, 60596 Frankfurt am Main, Germany; f.rothweiler@kinderkrebsstiftung-frankfurt.de (F.R.);; 3Institut für Rechtsmedizin, Klinikum der Goethe-Universität, 60596 Frankfurt am Main, Germany; zehner@em.uni-frankfurt.de; 4Leibniz-Institute Deutsche Sammlung für Mikroorganismen und Zellkulturen GmbH, 38124 Braunschweig, Germany; wdi@dsmz.de; 5Lonza Biologics, Slough SL1 4DX, UK; andy.racher@lonza.com

**Keywords:** cell line, authentication, cancer, mass spectrometry, isogenic

## Abstract

The use of cell lines in research can be affected by cell line misidentification. Short tandem repeat (STR) analysis is an effective method, and the gold standard, for the identification of the genetic origin of a cell line, but methods that allow the discrimination between cell lines of the same genetic origin are lacking. Here, we use intact cell MALDI-ToF mass spectrometry analysis, routinely used for the identification of bacteria in clinical diagnostic procedures, for the authentication of a set of cell lines consisting of three parental neuroblastoma cell lines (IMR-5, IMR-32 and UKF-NB-3) and eleven drug-adapted sublines. Principal component analysis (PCA) of intact-cell MALDI-ToF mass spectrometry data revealed clear differences between most, but not all, of the investigated cell lines. Mass spectrometry whole-cell fingerprints enabled the separation of IMR-32 and its clonal subline IMR-5. Sublines that had been adapted to closely related drugs, for example, the cisplatin- and oxaliplatin-resistant UKF-NB-3 sublines and the vincristine- and vinblastine-adapted IMR-5 sublines, also displayed clearly distinctive patterns. In conclusion, intact whole-cell MALDI-ToF mass spectrometry has the potential to be further developed into an authentication method for mammalian cells of a common genetic origin.

## 1. Introduction

Cell line misidentification has been a major issue since the first mammalian cell lines were established [1,2]. HeLa, the first human cancer cell line, was established in 1951 [3], and the first HeLa cell contamination of cell lines from nonhuman species was described in the early 1960s [4,5]. For example, in 1966, 18 cell lines, reported to be derived from different origins, were shown actually to be HeLa cells [6]. In two later studies, 16% [7] and 18% [8] of all cell lines submitted to cell culture depositories were reported to be misidentified. Short tandem repeat (STR) loci are highly informative polymorphic markers in the human genome. STR typing has become the gold standard to ensure the quality and authenticity of human cell lines in the scientific community, resulting in the establishment of guidelines for the authentication of cell lines, including the setup of databases with reference STR profiles [2,9,10,11,12,13]. Hence, reliable methods for the authentication of cell lines based on their genetic origin are available, although recent reports demonstrate that cell line misidentification remains an issue [14,15,16,17,18].

While there are established methods for the determination of the genetic origin of cell lines, methods for the discrimination of cell lines derived from the same genetic origin (isogenic cell lines), such as clonal sublines, different cell lines derived from the same organism, or sub-cell lines that were adapted to growth under specific conditions are currently under development. Intact-cell MALDI-ToF mass spectrometry is routinely used for the identification of bacteria in clinical diagnostic procedures [19] and has been used for the characterisation of various mammalian cell lines [20,21,22,23,24,25,26,27,28,29,30].

One group of cell lines where STR methods are not able to distinguish between them is drug-adapted cancer cell lines, which are used as in vitro models of acquired drug resistance. The ATP-binding cassette (ABC) transporters ABCB1 (also known as P-glycoprotein or MDR1) and ABCC1 (also known as MRP1), major mediators of drug resistance in cancer, were discovered in drug-adapted cell lines [31,32]. Moreover, drug-adapted cancer cell lines have been used by various research groups to identify and investigate clinically relevant resistance mechanisms to targeted and cytotoxic anticancer drugs (e.g., [33,34,35,36,37,38,39,40,41,42,43,44,45,46,47,48,49]). Here, we demonstrate that an intact-cell MALDI-ToF mass spectrometry approach [29] can in principle differentiate between parental cell lines and their drug-resistant sublines. This is a crucial first step towards the establishment of a methodology, including a database, for the reliable authentication of cell lines of a common genetic origin.

## 2. Results

### 2.1. Cell Line Authentication by Short Tandem Repeat (STR) Profiling

STR DNA typing with a detection limit of 1:10^5^ cells [50] did not indicate contamination of the study cell lines with mouse, rat, Chinese hamster or Syrian hamster cells. The cell lines were further authenticated using an extended set of STR markers (loci: D13S317, D16S539, D18S51, D19S433, D2S1338, D3S1358, D5S818, D7S820, D8S1179, CSF1PO, Amel, FGA, Penta D, Penta E, D21S11, TH01, TPOX, vWA, Amel), which is beyond the requirement of the standard of eight STR markers for unique identification [12] and has a matching probability of 1 in 2 × 10^17^ in Caucasian- and Hispanic-American populations. The genetic origin of all cell lines was successfully confirmed (Appendix A). IMR-5 is a clonal subline of IMR-32 [51] and only differed at one locus (PentaE) from IMR-32 (Appendix A).

Four of the eleven drug-adapted sublines displayed identical STR profiles to the respective parental cell lines (Figure 1, Appendix A). The other sublines showed one to four changes. Hence, all sublines displayed a >80% match with the respective parental cell lines, which is regarded as the threshold for authenticity [12]. In IMR-5^r^DACARB^40^, IMR-5^r^VINB^20^ and IMR-32^r^CARBO^1000^, the amelogenin STR locus indicated that the Y chromosome was lost and replaced by an X chromosome (Figure 1, Appendix A), which is a reported phenomenon in cell lines [52,53,54,55,56,57].

The investigated cell lines had previously been adapted to DNA-damaging agents including the platinum drugs cisplatin, carboplatin and oxaliplatin, the alkylating agents dacarbazine and melphalan, the topoisomerase I inhibitor topotecan, and antimitotic tubulin-binding agents including docetaxel, vinblastine and vincristine. The potential of these agents to affect STR profiles might have been expected to be higher for agents that cause direct DNA damage than for those that primarily inhibit cell division. However, the vinblastine-resistant IMR-5 subline IMR-5^r^VINB^20^ was the only cell line that displayed four STR changes, one of which was the exchange of the Y for an X chromosome. Three of the four sublines, which did not differ in their STR profiles from the parental cell lines, were sublines adapted to the DNA-damaging agents cisplatin, oxaliplatin and melphalan (Figure 1, Appendix A). These data suggest that DNA-damaging agents do not commonly affect STR integrity.

### 2.2. Intact-Cell MALDI-ToF Mass Spectrometry Analysis

Representative intact-cell MALDI-ToF mass spectrometry analysis spectra are provided in Figure 2, Figure 3 and Figure 4. Thorough visual comparison of the spectra enables the determination of characteristic differences in the profiles. Some changes are highlighted in Appendix A.

Next, we investigated whether data analysis using principal component analysis (PCA) could discriminate between different cell lines. First, we looked at the parental cell lines IMR-32, IMR-5 and UKF-NB-3 using the comparisons PC1 vs. PC2, PC1 vs. PC3 and PC2 vs. PC3. The comparison PC2 vs. PC3 showed the largest separation (Figure 5A, Appendix A). Notably, the method enabled the discrimination of IMR-32 and its clonal subline IMR-5, indicating that intact-cell MALDI-ToF mass spectrometry analysis is in principle suited to differentiate between cell lines of a common genetic origin. Similar results were obtained for the comparisons of UKF-NB-3, IMR-32 and IMR-5 and their respective drug-adapted sublines (Figure 5B–D). The comparison PC2 vs. PC3 consistently resulted in the best separation (Figure 5A–D, Appendix A).

The MALDI-ToF intact-cell spectra and subsequent PCA approach separated the parental UKF-NB-3 cell line from its sublines adapted to cisplatin (UKF-NB-3^r^CDDP^1000^) and oxaliplatin (UKF-NB-3^r^OXALI^2000^). It further separated the two sublines that had been adapted to closely related platinum drugs (Figure 5B, Appendix A).

The analysis of a larger set of IMR-32 sublines also resulted in separation of individual cell lines (Figure 5C, Appendix A). The sublines adapted to carboplatin (IMR-32^r^CARBO^1000^), topotecan (IMR-32^r^TOPO^8^) and melphalan (IMR-32^r^MEL^500^) displayed clearly distinctive patterns. Only the oxaliplatin-resistant IMR-32 subline (IMR-32^r^OXALI^800^) was difficult to differentiate from the parental IMR-32 cells (Figure 5C, Appendix A).

In the drug-adapted IMR-5 sublines, the parental cell line, the vinblastine-adapted subline (IMR-5^r^VINB^20^) and the cisplatin-adapted (IMR-5^r^CDDP^1000^) subline formed clearly distinctive clusters (Figure 5D, Appendix A). The docetaxel-resistant (IMR-5^r^DOCE^20^), dacarbazine-resistant (IMR-5^r^DACARB^40^) and vincristine-resistant (IMR-5^r^VCR^10^) sublines were harder to distinguish, although they also formed individual clusters (Figure 5D, Appendix A).

## 3. Discussion

Cell line misauthentication remains an important problem in life sciences research [14,15,16,17,18]. While STR analysis has been established as an effective method for the identification of the genetic origin of a cell line [2,9,10,12,13], established methods for the discrimination of cell lines of a common genetic origin are lacking. Here, we performed an initial study based upon using intact-cell MALDI-ToF mass spectrometry analysis and subsequent PCA of the resulting spectra for the separation of cell line sets consisting of three parental cell lines and eleven sublines adapted to a range of different anticancer drugs.

Our STR analysis using an expanded set of STR markers, which has a matching probability of 1 in 2 × 10^17^ in Caucasian- and Hispanic-American populations, confirmed the identity of the investigated cell lines. Cell line adaptation to anticancer drugs had little to no effect on the STR profiles and did not interfere with the determination of the cell lines’ genetic origins, although the drugs that had been used included DNA-damaging agents such as the alkylating agents melphalan and dacarbazine and the platinum drugs cisplatin, carboplatin and oxaliplatin. This confirms previous findings, which had shown that the ovarian cancer cell line OVCAR8 and its doxorubicin-adapted subline OVCAR8/ADR still share a common STR profile [58].

Intact-cell MALDI-ToF mass spectrometry analysis that is routinely used for the identification of bacteria in clinical diagnostic procedures [19], and has been applied to the characterisation of various mammalian cell lines [20,21,22,23,24,25,26,27,28,29,30], is an alternative authentication method that has the potential to be further developed for the authentication of cell lines of the same genetic origin. Our initial results presented here show that intact-cell fingerprinting by MALDI-ToF mass spectrometry, followed by subsequent principal component analysis (PCA) reveals clear differences in the spectra between most of the investigated cell lines. This approach enabled the separation of the neuroblastoma cell line IMR-32 and its clonal subline IMR-5. In addition, sublines that had been adapted to closely related drugs, for example, the cisplatin- and oxaliplatin-resistant UKF-NB-3 sublines and the vincristine- and vinblastine-adapted IMR-5 sublines, displayed clearly distinctive patterns. Hence, the data presented here shows that intact-cell MALDI-ToF mass spectrometry has the potential to be further developed into an authentication method for mammalian cell lines derived from a common genetic origin.

We note that the initial approach developed here was not able to discriminate between all the cell lines that we investigated in our study. There was no clear separation of IMR-32 and its oxaliplatin-adapted subline and among the docetaxel-, dacarbazine-, and vincristine-resistant sublines of IMR-5. Although a combination of drug sensitivity data and intact whole-cell MALDI-ToF mass spectrometry would enable the correct authentication of these cell lines, a method that enables cell line identification without dependence on additional functional assays is much more desirable. Interestingly, sublines adapted to closely related drugs from the same class (e.g., the vincristine- and vinblastine-adapted IMR-5 sublines and the carboplatin- and oxaliplatin-adapted IMR-32 sublines) did not cluster closely together with regard to their mass spectrometry profiles. Further research will have to show whether this observation has functional implications.

When directly comparing the STR and intact-cell MALDI-ToF analysis, as stated above, the STR approach showed four of the eleven drug-adapted sublines displayed identical STR profiles to the respective parental cell lines and the other sublines showed one to four changes, and hence all sublines displayed a >80% match with the respective parental cell lines, regarded as the threshold for authenticity [12] and hence would be considered identical to the parental. On the other hand, using the intact-cell MALDI-ToF methodology allowed separation of IMR-32 and the IMR-5 sublines, whilst IMR-5 and UKF-NB-3 drug-adapted cell lines tended to show good separation. Thus, the MALDI-ToF approach does appear sensitive enough to differentiate between sublines that the STR profiling does not. As stated above and below, further work and optimisation needs to be undertaken to determine if intact-cell MALDI-ToF analyses can be used to differentiate a wider panel of cell lines and subcell lines that can complement STR analyses and provide a rapid method for investigators to probe the identity and authenticity of cell lines.

In conclusion, intact-cell MALDI-ToF mass spectrometry is a promising technique for the authentication of mammalian cells of a common genetic origin, but further development is required before it can be used as a reliable, generally applicable method, either on its own or (which seems to be more likely for the foreseeable future) in combination with other established methods, in particular STR typing. This approach represents a rapid method of authenticating cell lines that could complement traditional STR profiling. In developing this approach to authentication, we recommend that a minimum of three replicate analyses be undertaken for each sample to account for technical variation, and ideally three biological samples would also be analysed. We also suggest that a bank of quality controls consisting of historical fingerprints of authenticated cell lines be curated alongside frozen samples of authenticated cells that could be analysed alongside naive samples to account for any day-to-day variation in analyses and to give further confidence to identification of unknown cell lines. Further research will have to (1) establish that the methodology is able to identify unknown cell lines using statistically robust validation approaches and by masking the identification of cell lines to experimentalists, (2) show the extent to which cross-contaminations can be detected based on mass spectrometry profiles, (3) investigate the potential influence of cell culture conditions on mass spectrometry profiles and (4) establish databases with cell line profiles and quality standards for wider application by the community. Notably, methods for the authentication of cell lines of the same origin will also be important for areas beyond cancer research. For example, mouse strains are often derived from inbred strains and can also not be distinguished by STR [56,59].

## 4. Materials and Methods

### 4.1. Cell Lines

The MYCN-amplified neuroblastoma cell line UKF-NB-3 was established from a bone marrow metastasis derived from a stage 4 neuroblastoma patient [60]. The MYCN-amplified neuroblastoma cell line IMR-32 was obtained from ATCC (Manassas, VA, USA), and its clonal subline IMR-5 [51] was kindly provided by Dr Angelika Eggert (Universität Duisburg-Essen, Essen, Germany).

UKF-NB-3 sublines with acquired resistance to cisplatin (UKF-NB-3^r^CDDP^1000^) or oxaliplatin (UKF-NB-3^r^OXALI^2000^), IMR-32 sublines adapted to carboplatin (IMR-32^r^CARBO^1000^), melphalan (IMR-32^r^MEL^500^), oxaliplatin (IMR-32^r^OXALI^800^) or topotecan (IMR-32^r^TOPO^8^), and IMR-5 sublines adapted to cisplatin (IMR-5^r^CDDP^1000^), dacarbazine (IMR-5^r^DACARB^40^), docetaxel (IMR-5^r^DOCE^20^), vinblastine (IMR-5^r^VINB^20^) or vincristine (IMR-5^r^VCR^10^) were established by continuous exposure to stepwise increasing drug concentrations, as previously described [38,60,61] and derived from the Resistant Cancer Cell Line (RCCL) collection (https://research.kent.ac.uk/ibc/the-resistant-cancer-cell-line-rccl-collection/).

All cell lines were propagated in Iscove’s modified Dulbecco’s medium (IMDM) supplemented with 10% foetal calf serum (FCS), 100 IU/mL penicillin and 100 µg/mL streptomycin at 37 °C. Cells were routinely tested for mycoplasma contamination.

### 4.2. Cell Line Authentication by Short Tandem Repeat (STR) Profiling

Cell lines were authenticated with the current STR typing technique of the DSMZ (Deutsche Sammlung von Mikroorganismen und Zellkulturen GmbH, Braunschweig, Germany), which, according to the STR Typing Standard (ASN-0002: Human cell line authentication: STR profile standardization), uses an expanded set of STR markers that goes beyond the standard set of eight markers (loci: D13S317, D16S539, D18S51, D19S433, D2S1338, D3S1358, D5S818, D7S820, D8S1179, CSF1PO, FGA, Penta D, Penta E, D21S11, TH01, TPOX, vWA, Amel).

Amplification was performed in DNase- and RNase-free vials of 0.2 mL in an i-Cycler (BioRad, Puchheim, Germany). Per sample, the master mix contained 10 pmol total primer solution (1 µL of a 10 µM solution), 2.5 µL 10 x hot start PCR buffer (any supplier), 1 µL dNTP solution (5 µM), 0.2 µL hot start Taq polymerase solution (1 unit, any supplier), 19.5 µL distilled water and 1 µL genomic DNA solution (10 ng/µL). The genomic DNA was added last and independent from the master mix. The PCR programme was carried out as follows: 95 °C for 3 min, 1 repeat; 94 °C for 30 s, 57 °C for 30 s, 72 °C for 45 s, 30 repeats; 60 °C for 15 min, 1 repeat.

For fragment detection and generation of allelic STR lists, aliquots of 1 µL of PCR products were combined with 0.25 µL of an internal size standard (Size standard kit 400, Beckman-Coulter, High Wycombe, UK) in a total volume of 30 µL of sample loading solution on a microtiter plate. The samples were loaded automatically and analysed by the capillary electrophoresis system CEQ 8000 (Beckman-Coulter) using established fragment analysis parameters. Generated STR profiles were submitted to search within a reference STR database of cell lines [11]. Additionally, all cell line samples were tested for the presence of mitochondrial DNA sequences from mouse, rat, Chinese hamster and Syrian hamster [50].

### 4.3. Intact-Cell MALDI-ToF Mass Spectrometry Analysis

The intact-cell MALDI-ToF mass spectrometry analysis was performed essentially as previously described [29]. 6.25 × 10^4^ viable cells were transferred to a 96-well plate format rack, centrifuged for 5 min at 960 rcf, and the supernatant removed. After washing by resuspension and centrifugation in 0.5 mL of PBS and in 0.35 M sucrose, cell pellets were stored at –80 °C until analysed.

20 mg/mL (saturated) solution of sinapinic acid was dissolved in matrix buffer (40% acetonitrile, 0.06% TFA (trifluoroacetic acid)) and sonicated in a water bath for 15 min before centrifugation. After removal from –80 °C storage, cells were allowed to equilibrate to room temperature for 15 min prior to resuspension in 50 µl of sinapinic acid solution and incubation at 4 °C for 3 h. Then, 1 µl of each sample was spotted onto a 384 MTP ground steel MALDI-ToF plate (Bruker, Coventry, UK) and air-dried, before the plate was placed into the MALDI-ToF mass spectrometry instrument (Bruker Ultraflex).

Spectra were collected using the following MALDI-ToF ms instrument settings: laser frequency: 20 Hz; polarity +ve; ion sources: 1. 20 kV, 2. 17.25 kV, lens 5.0 kV; gating mode: maximum strength; suppress @ 4000 Da; pulsed ion extraction 550 nS; range 5000–60,000 Da; sample rate 0.1 Gs/s; resolution enhanced 100 mV electronic gain; smooth high. For each sample, 100 shots were summed and saved. In all cases, triplicate (*n* = 3) biological cultures were analysed, each culture analysed by triplicate technical analyses of the samples.

Data files were exported from the Bruker Flex Analysis software in ASCII format for preprocessing in MATLAB (Version R2017a for Windows) prior to the application of the principal component analysis as described in [29]. The following preprocessing steps were applied to the spectra prior to presenting to the Partial least squares discriminant analysis (PLS-DA) algorithm: 1. Resampling (up-sample to 50,000 to account for slight differences in *m/z* vector); 2. Baseline Correction (removes the effect of noise introduced by the matrix); 3. Filtering (Application of Savitzky-Golay filter to smooth the signal); 4. Alignment (automatically select peak and align spectra based on height and over-segmentation filters); 5. Quality Control (outlier detection and removal of “unusual spectra”); 6. Normalisation (normalisation of area under curve). Details for all steps have been outlined previously in [29]. Resampling of the mass spectrometry profiles was performed using the “msresample” function from the MATLAB Bioinformatics Toolbox (http://tinyurl.com/msresample). In addition, a baseline correction was performed using the “msbackadj” function in the MATLAB Bioinformatics Toolbox (http://tinyurl.com/msbackadj). Peak alignment was performed using the “msalign” function from the MATLAB Bioinformatics Toolbox (http://tinyurl.com/msalign). Finally, data was normalised using the “msnorm” function from the MATLAB Bioinformatics Toolbox (http://tinyurl.com/msnormal), as described in [29].

## Figures and Tables

**Figure 1 cells-08-01194-f001:**
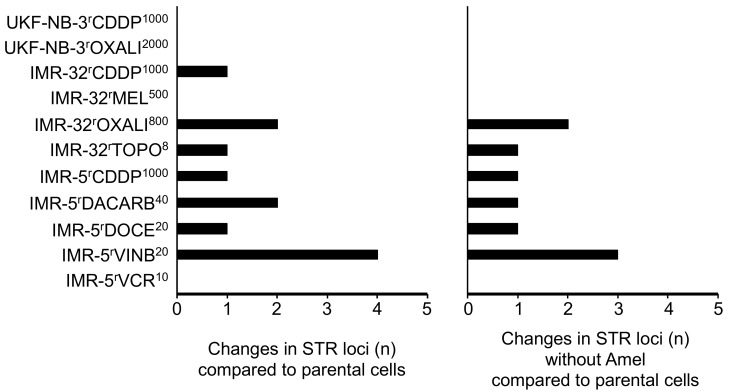
Numbers of STR (short tandem repeat) loci in which drug-adapted sublines differ from the respective parental cell lines with or without the amelogenin locus (Amel) that discriminates between the X and the Y chromosome.

**Figure 2 cells-08-01194-f002:**
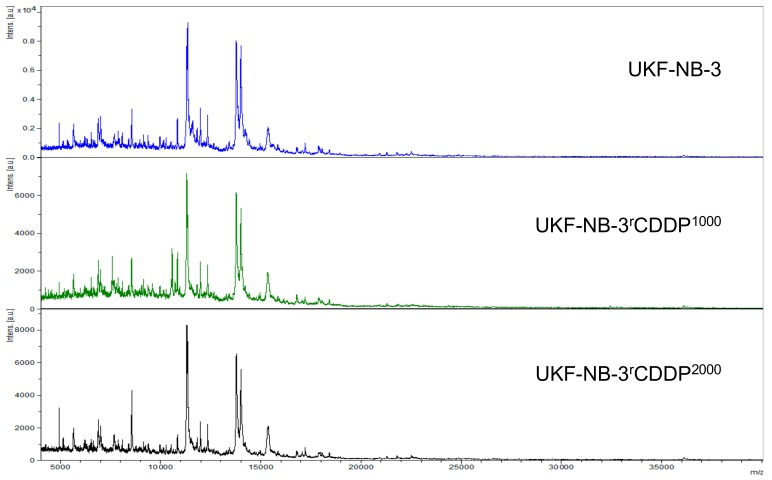
Representative intact-cell MALDI-ToF mass spectrometry analysis spectra of the cell line UKF-NB-3 and its drug-adapted sublines.

**Figure 3 cells-08-01194-f003:**
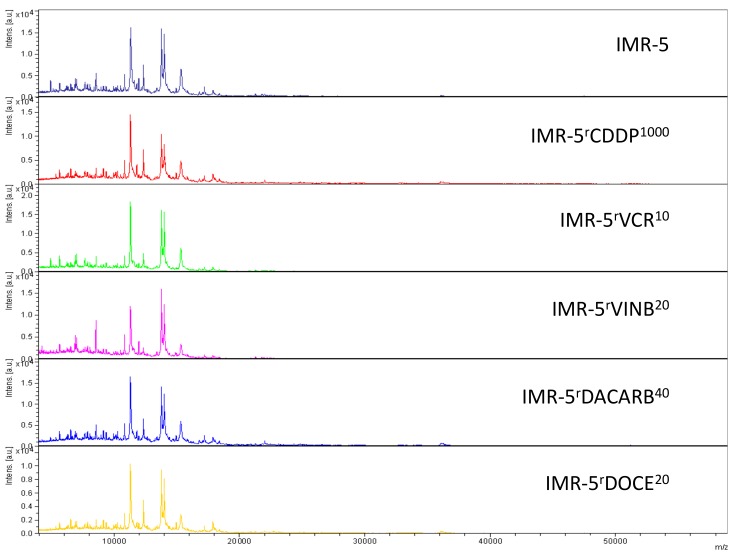
Representative intact-cell MALDI-ToF mass spectrometry analysis spectra of the cell line IMR-5 and its drug-adapted sublines.

**Figure 4 cells-08-01194-f004:**
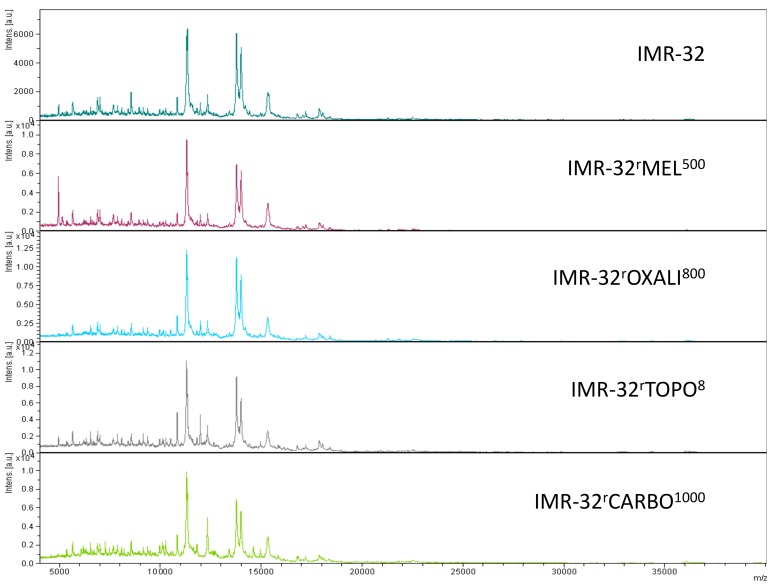
Representative intact-cell MALDI-ToF mass spectrometry analysis spectra of the cell line IMR-32 and its drug-adapted sublines.

**Figure 5 cells-08-01194-f005:**
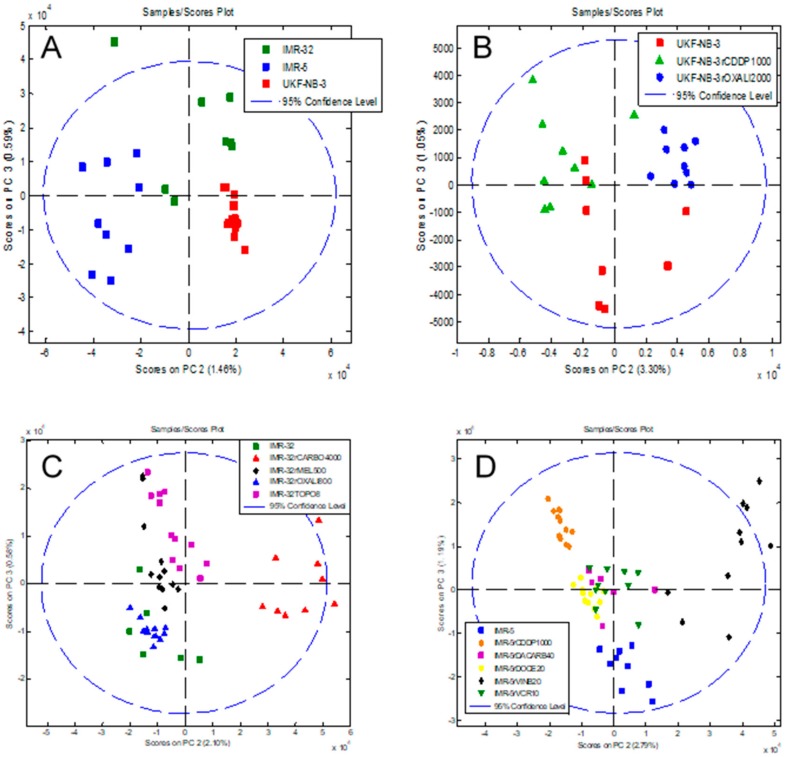
Principal component analysis (PCA) of the intact-cell MALDI-ToF mass spectrometry spectra data derived from the different cell lines, showing a plot of component PC2 vs. PC3 that allows discrimination of different cell subtypes and resulted in the best discrimination of the cell lines. **(A)** Parental cell lines IMR-32, IMR-5 and UKF-NB-3. The comparisons PC1 vs. PC2 and PC1 vs. PC3 are presented in Appendix A. **(B)** UKF-NB-3 and its drug-adapted sublines. The comparisons PC1 vs. PC2 and PC1 vs. PC3 are presented in Appendix A. **(C)** IMR-32 and its drug-adapted sublines. The comparisons PC1 vs. PC2 and PC1 vs. PC3 are presented in Appendix A. **(D)** IMR-5 and its drug-adapted sublines. The comparisons PC1 vs. PC2 and PC1 vs. PC3 are presented in Appendix A. In each case, the individual data points represent the analysis of a single biological cell line. For each of the host or drug-adapted cell lines, three biological cultures were analysed in triplicate by intact-cell MALDI-ToF and thus there are nine data points for each cell line.

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
