# Peer review of "Intact-Cell MALDI-ToF Mass Spectrometry for the Authentication of Drug-Adapted Cancer Cell Lines"

_cells, 2019, doi:10.3390/cells8101194_

Round 1
Reviewer 1 Report
The manuscript “Intact whole cell MALDI-ToF mass spectrometry for the authentication of drug-adapted cancer cell lines” by Povey et al. is, a well-organized and targeted paper by an established group. The specific paper targets the question whether a MALDI ToF-MS/MS method, based on the analysis of intact cells, can be used to separate between three parental neuroblastoma cell lines and eleven drug-adapted cell lines. There are some points that need the authors’ attention:
Some confusion arises from the fact, that the methods part indicates a PLS-DA (partial least square discriminant analysis), but the data presented are seemingly from a PCA (principal component analysis). Clarification is needed and if a PLS-DA was performed, the actual results for the separation are needed. With the data set a discriminant analysis should be performed to proof the discriminating power of the model. A validation step (preferably leave-one-out formalism) should be used to verify the model. The major compounds responsible for the separation should be identified and major metabolic pathways involved should be added. Possible gene-dependent metabolic pathway regulation should be added to the discussion. Seemingly an outlier algorithm was applied. The exact samples, that were treated as outliers and were subsequently not included into the data sets, needs to be specified. Maybe I missed it, but I could not find anything about the actual used samples. How many samples were there for each cell line, how many repeats were analysed a) for each well b) for different wells from the same cell line, were all cell cultivated at the same time? For figure 5 there somehow needs to be clarified, what all the blue spots represent (sample number, biological and/or sample repeat etc). This applies to all coloured dots in fig 5 and to figure 6,7 and 8.Author Response
The manuscript “Intact whole cell MALDI-ToF mass spectrometry for the authentication of drug-adapted cancer cell lines” by Povey et al. is, a well-organized and targeted paper by an established group. The specific paper targets the question whether a MALDI ToF-MS/MS method, based on the analysis of intact cells, can be used to separate between three parental neuroblastoma cell lines and eleven drug-adapted cell lines.
Authors’ response:
We thank the reviewer for these comments, who clearly recognises the potential of this approach based upon this proof-of-concept study.
There are some points that need the authors’ attention: Some confusion arises from the fact, that the methods part indicates a PLS-DA (partial least square discriminant analysis), but the data presented are seemingly from a PCA (principal component analysis). Clarification is needed and if a PLS-DA was performed, the actual results for the separation are needed. With the data set a discriminant analysis should be performed to proof the discriminating power of the model. A validation step (preferably leave-one-out formalism) should be used to verify the model.
Authors’ response:
We thank the reviewer for highlighting this. At this stage, and due to the number of datasets we have, we have just undertaken PCA analysis to show how intact MALDI-ToF analysis can be used to discriminate between cell lines as previously reported by others in different fields and referred to in the text. We have removed the reference to PLS-DA and entirely agree with the reviewer that the next logical step to follow this proof-of-concept study is to generate larger datasets so we can use leave-one-out cross validation in a PLS-DA (or similar) model to validate the model for the prediction of cell line identity and thus authentication. This would be accompanied by using and developing standard fingerprints and databases of authenticated cell lines alongside the generation of biological standard samples of known cell types to run with unknown samples (as suggested by reviewer 3 as well).
The major compounds responsible for the separation should be identified and major metabolic pathways involved should be added. Possible gene-dependent metabolic pathway regulation should be added to the discussion.
Authors’ response:
It is not possible to identify any gene dependent pathways or individual genes from the MALDI-ToF fingerprints alone. We agree that this would be very interesting and inform on any biological differences between cell lines and the mechanisms that might underpin difference between sub-types. However, this would require a different proteomic approach than the generation of the intact MALDI-ToF data generated in this study.
Seemingly an outlier algorithm was applied. The exact samples, that were treated as outliers and were subsequently not included into the data sets, needs to be specified.
Maybe I missed it, but I could not find anything about the actual used samples. How many samples were there for each cell line, how many repeats were analysed a) for each well b) for different wells from the same cell line, were all cell cultivated at the same time?
Authors’ response:
We apologise that this was not clear. We have stated the number of replicates in the analysis in the methods and figure legends (3 biological replicates for each cell line with each replicate analysed in technical triplicate). Outlier identification, quality control and normalisation details are provided in reference [29]. We have now added this detail to the manuscript. Not all cells were cultivated at the same time but at different times.
For figure 5 there somehow needs to be clarified, what all the blue spots represent (sample number, biological and/or sample repeat etc). This applies to all coloured dots in fig 5 and to figure 6,7 and 8.
Authors’ response:
We apologise for not providing sufficient explanation and thank the reviewer for highlighting this. As outlined in response to reviewer 3 as well, we have combined the PCA plot figures into one new figure with all the PCA plots. This new figure (Figure 5), contains the previous figures 5-8 as a multi-panel5A-5D figure and the figure legend has been expanded to include an explanation of what all the data points represent.
Reviewer 2 Report
The manuscript describes the MALDI-TOF MS analysis of intact whole cells for the authentication of drug-adapted cancer cell lines. MALDI-TOF MS analysis has been extensively used for the identification of bacteria in clinical diagnostic procedures. The adaptation of MALDI-TOF MS for the intact whole cells would therefore be very promising. The authors concluded that MALDI-TOF MS for authentication of mammalian cells is a promising technique, but further requirements are needed. Even though the conclusion is not conclusive, the results would be very informative for those are using MALDI-TOF MS for the cell identification.
One major thing that needs to be addressed before publication is the comparison of STR results with MALDI-TOF MS results. More solid comparison would improve the quality of the manuscript.
The followings show minor correction points.
Line 27. IMR-5, IMR-32, UKF-NB-3 -> IMR-5, IMR-32, and UKF-NB-3
Figure 5. The legends should have different shapes, so they can be distinguishable in B/W colors.
Author Response
The manuscript describes the MALDI-TOF MS analysis of intact whole cells for the authentication of drug-adapted cancer cell lines. MALDI-TOF MS analysis has been extensively used for the identification of bacteria in clinical diagnostic procedures. The adaptation of MALDI-TOF MS for the intact whole cells would therefore be very promising. The authors concluded that MALDI-TOF MS for authentication of mammalian cells is a promising technique, but further requirements are needed. Even though the conclusion is not conclusive, the results would be very informative for those are using MALDI-TOF MS for the cell identification.
We thank the reviewer for these positive comments.
One major thing that needs to be addressed before publication is the comparison of STR results with MALDI-TOF MS results. More solid comparison would improve the quality of the manuscript.
Authors’ response:
We have expanded the comparison of the STR and MALDI-ToF results in the discussion section as requested, adding a new paragraph that addresses this.
The followings show minor correction points.
Line 27. IMR-5, IMR-32, UKF-NB-3 -> IMR-5, IMR-32, and UKF-NB-3
Authors’ response:
We have corrected this.
Figure 5.
The legends should have different shapes, so they can be distinguishable in B/W colors.
Authors’ response:
We have investigated this and it is very hard to distinguish between different shapes in B/W. We anticipate the manuscript being published in colour which gives the best distinction between the different cell lines/sub-types.
Reviewer 3 Report
In Povey et al., a MALDI-TOF based approach for discerning genetically identical cell lines is presented. The problem is very real for centers/institutions/departments/companies dealing with a large number of isogenic cell lines. The approach is sound in reasoning and has been used by others for cell profiling. However, the paper requires major modifications for publication.
Major:
The greatest test of the method is to mask the identities of cells and then compare to reference spectra generated from cells with known identity as well as positive controls included within an experiment. The authors reveal that not all populations can be discerned using their approach, but this test should be performed at least for the sub-types which can be differentiated.
The day-to-day variability needs to be assessed since this method will presumably be performed routinely over time. Quality controls would also be warranted (cells of confirmed identity frozen in aliquots and analyzed on each day).
There are too many figures. The PCA plots could be reduced in size and included in a multi-panel figure. Though I do appreciate the inclusion of raw spectra, the highlighted differences in the spectra should be in the manuscript not the supplement.
The methods require improvement. How many replicates were included in each analysis? How many ions were detected in each sample? "Quality Control (Outlier detection and removal of “unusual spectra”)" What are unusual spectra? How was the normalization performed?
m/z should be m/z (minor test error but please correct).
The version of MATLAB needs to be reported.
Minor:
A heatmap showing the cell types and their sub-types with clustering is a necessary comparison to the PCA plots. It is very likely that a heatmap will disallow distinguishing between groups; however, many groups rely upon such figures to detect and summarize differences.
If this procedure is to be used routinely, a recommended number of replicates required would be a good addition.
Author Response
In Povey et al., a MALDI-TOF based approach for discerning genetically identical cell lines is presented. The problem is very real for centers/institutions/departments/companies dealing with a large number of isogenic cell lines. The approach is sound in reasoning and has been used by others for cell profiling. However, the paper requires major modifications for publication.
Authors’ response:
We thank the reviewer for these comments and are pleased the reviewer has recognised the potential of this approach.
Major:
The greatest test of the method is to mask the identities of cells and then compare to reference spectra generated from cells with known identity as well as positive controls included within an experiment. The authors reveal that not all populations can be discerned using their approach, but this test should be performed at least for the sub-types which can be differentiated.
Authors’ response:
Indeed, the ultimate test will be to analyse unknown or naïve cell lines and determine if this can accurately identify/authenticate these. This is required further work in developing the technology and will be an important aspect of the next steps in its development. We have highlighted this in the conclusions paragraph.
The day-to-day variability needs to be assessed since this method will presumably be performed routinely over time. Quality controls would also be warranted (cells of confirmed identity frozen in aliquots and analyzed on each day).
Authors’ response:
We thank the reviewer for highlighting this. Indeed, if this was to be used further quality controls to validate experiments would be required and we have added this important point to the manuscript. The method has excellent day-to-day reproducibility as we and others have previously shown and we refer to this in the text to address this point.
There are too many figures. The PCA plots could be reduced in size and included in a multi-panel figure. Though I do appreciate the inclusion of raw spectra, the highlighted differences in the spectra should be in the manuscript not the supplement.
Authors’ response:
Thank you to the reviewer for this suggestion. We have combined the PCA plot figures into one new figure with all the PCA plots. This new figure (Figure 5), contains the previous figures 5-8 as a multi-panel5A-5D figure.
The methods require improvement. How many replicates were included in each analysis? How many ions were detected in each sample? "Quality Control (Outlier detection and removal of “unusual spectra”)" What are unusual spectra? How was the normalization performed?
Authors’ response:
We apologise that this was not clear. We have stated the number of replicates in the analysis in the methods and figure legends (3 biological replicates for each cell line with each replicate analysed in technical triplicate). Quality control and normalisation details are provided in reference [29} and we have now added this detail to the manuscript.
m/z should be m/z (minor test error but please correct).
Authors’ response:
We are not sure what the reviewer refers to here? We have checked all m/z and these appear correct.
The version of MATLAB needs to be reported.
Authors’ response:
Matlab R2017a
Minor:
A heatmap showing the cell types and their sub-types with clustering is a necessary comparison to the PCA plots. It is very likely that a heatmap will disallow distinguishing between groups; however, many groups rely upon such figures to detect and summarize differences.
Authors’ response:
We thank the reviewer for this suggestion. However, as the reviewer states, a heat map does not show any meaningful distinguishing between the groups and in an authentication sense does not aid the user. Indeed, if the approach is to be used for authentication we would need to develop a more user-friendly interface with the data that compared the fingerprint of an unknown with that of a database and gave an indication of the match/authentication and confidence in that match.
If this procedure is to be used routinely, a recommended number of replicates required would be a good addition.
Authors’ response:
We recommend 3 replicates for those wishing to undertake such analysis routinely. We have added this to the text.
Round 2
Reviewer 1 Report
Based on the changes made in the manuscript and the answers formulated in the rebuttal letter all issues are cleared.
Reviewer 3 Report
In response to the Author's replies, I appreciate your answers to my major and minor concerns. I believe that the textual responses and reformatting of the figures are satisfactory. The minor detail of the format in regard to mass-to-charge ratio is that it should be italicised (m/z). In addition, there are minor English errors in the added text, which are minor in nature. For these reasons, I recommend the publication for publication with minor revision.